# Current Utilization and Research Status of Herbal Medicine Sipjeondaebotang for Anemia: A Scoping Review

**DOI:** 10.3390/ph17091192

**Published:** 2024-09-10

**Authors:** Gyeongmuk Kim, Han-Gyul Lee, Won Jung Ha, Seungwon Kwon

**Affiliations:** 1Department of Clinical Korean Medicine, Graduate School, Kyung Hee University, Seoul 02447, Republic of Korea; kkm156@naver.com (G.K.); new_mon@naver.com (W.J.H.); 2Department of Cardiology and Neurology, Kyung Hee University College of Korean Medicine, Kyung Hee University Medical Center, Seoul 02447, Republic of Korea; gyulee0614@hanmail.net

**Keywords:** anemia, Sipjeondaebotang, Shiquan Dabu tang, Juzentaihoto, hemoglobin, hematocrit

## Abstract

Background/Objectives: Anemia is a global health issue affecting diverse populations, particularly older adults, and conventional treatments often show limited efficacy. This study aimed to evaluate the utilization and effectiveness of Sipjeondaebotang (SDT), a prescription drug used in traditional East Asian medicine, in treating various types of anemia. Methods: A scoping review was conducted following Arksey and O’Malley’s framework and PRISMA-ScR guidelines. Six electronic databases were searched for clinical studies on SDT, while focusing on human participants and excluding animal and cellular studies. Sixteen studies, including nine randomized controlled trials, two controlled clinical trials, two case series, and three case reports, involving 863 participants, were analyzed. These studies were primarily conducted in China, Korea, and Japan. Results: According to the analysis, SDT improved hemoglobin levels across all types of anemia studied, with all controlled studies showing significant improvements compared with the control groups. Additionally, SDT reduced blood loss, improved recovery times, and decreased transfusion requirements in patients with post-operative anemia, with lower adverse event rates than those in the control groups. These findings suggest that SDT may enhance hematological parameters and improve overall patient outcomes. Conclusions: In conclusion, SDT may be an effective treatment for anemia that improves hemoglobin levels and patient outcomes. However, further high-quality, large-scale studies are necessary to standardize SDT prescriptions, confirm the optimal treatment duration, and validate its efficacy and safety across different anemia types.

## 1. Introduction

Anemia is a condition characterized by a decrease in the number of red blood cells (RBCs) or hemoglobin levels, leading to an inadequate oxygen supply to body tissues [1,2]. Anemia is a global public health problem that affects people of all ages in both developing and developed countries [3,4]. Especially in individuals aged ≥ 65 years, anemia significantly contributes to morbidity and mortality and affects the quality of life. In the context of global aging, anemia is a burden on individuals, society, and healthcare providers [5,6].

Anemia is classified based on its etiology, including types such as iron deficiency anemia, which results from defects in the differentiation and maturation of erythroid cells; vitamin deficiency anemia, caused by deficiencies in vitamin B12 and folic acid; anemia due to bleeding; hemolytic anemia; aplastic anemia; chronic disease-related anemia due to conditions such as renal failure or malignant tumors; and hereditary diseases such as sickle cell anemia [7].

Iron deficiency anemia is treated with oral or intravenous iron supplements such as iron sucrose [8]. Pernicious anemia due to vitamin B12 or folate deficiency is managed with vitamin B12 and folate supplements [9]. Anemia in chronic kidney disease (CKD) patients is treated with iron supplements and erythropoietin receptor activators [10]. These conventional Western treatments improve hemoglobin levels and alleviate symptoms. However, these treatments may not be effective in all cases. In the case of anemia in CKD patients, even when erythropoietin receptor activators are used at appropriate doses, 5–10% of CKD patients show insufficient clinical response [11]. Additionally, for the treatment of iron deficiency anemia, only approximately 10% of orally administered iron supplements are absorbed in the intestine, indicating several clinical limitations [12]. 

In East Asia, herbal medicine prescriptions from traditional East Asian medicine (TEAM) have been orally administered to manage clinical symptoms associated with anemia, which is recognized as a clinical syndrome known as “blood deficiency” in TEAM theory [13,14]. Herbal prescriptions for the treatment of this condition have been developed over time, and their indications and effects have been documented in medical texts and clinical case studies [15,16,17].

Among these, Sipjeondaebotang (SDT; Shiqua Dabu Tang in Chinese and Juzentaihoto in Japanese) is a prescription composed of 10 herbal medicines (Table 1). SDT is a herbal prescription that tonifies blood deficiency. In a previous systematic review of anemia [17], it was noted that Astragali Radix (Root of *Astragalus membranaceus* Bunge), Atractylodis Rhizoma Alba (Rhizome of *Atractylodis Rhizoma* Alba), and Angelicae Gigantis Radix (Root of *Angelica gigas* Nakai) were the most commonly used herbal medicines in the analyzed studies. Since SDT contains these herbal medicines, it raises the possibility of improving anemia by tonifying the blood effects of SDT. Recently, clinical reports on the use of SDT for anemia were published. Chen et al. and Ai et al. conducted randomized controlled trials (RCT) on SDT in patients with post-operative anemia and reported that the combination of SDT and conventional therapy was more beneficial than conventional therapy alone in improving anemia [18,19]. However, there is limited comprehensive clinical information on the effects of SDT on anemia. 

To clarify the effects and indications of SDT for anemia, we reviewed all clinical reports and reviews on SDT for anemia. This review aimed to evaluate comprehensively the current evidence supporting the use of SDT in treating anemia, determine its significance in clinical practice and identify knowledge gaps. This review will help establish a clearer understanding of SDT clinical efficacy, guiding future research to enhance its clinical application.

## 2. Results

### 2.1. Literature Search and Selection Process

Seventy documents published up to 30 June 2024 were identified. Finally, 16 studies [18,19,20,21,22,23,24,25,26,27,28,29,30,31,32,33] were selected using a literature selection process (Figure 1). A total of 22 duplicate studies were excluded from the analysis. In addition, 31 studies were excluded based on their titles and abstracts because they were not directly related to SDT or anemia, were not conducted in human patients, or were not related to herbal medicines. Following a professional review, one study was excluded because the original text could not be located.

### 2.2. General Characteristics of the Identified Literature

The relevant literature was published over the past 26 years (1999–2024). Except for one Chinese paper published in 1999 [20], all publications date from 2000 onward (Table 2).

### 2.3. Study Designs

Of the 16 studies reviewed, five were observational studies (31.25%) [20,21,24,30,31] and 11 were interventional studies (68.75%) [18,19,22,23,25,26,27,28,29,32,33]. Among the observational studies, three were case reports [24,30,31] and two were case series [20,21]. The interventional studies included nine RCTs [18,19,25,26,27,28,29,32,33] and two controlled clinical trials [22,23].

Of the nine RCTs, six focused on interventions for anemia resulting from post-operative bleeding, all of which were related to orthopedic surgeries [18,19,27,29,32,33]. Two studies addressed patients receiving erythropoietin for CKD anemia [25,26], while one study investigated anemia due to gastrointestinal bleeding [28]. 

One of the two case series examined hemolytic anemia after interferon/ribavirin therapy in patients with hepatitis C [21]. The three case reports addressed treatments for CKD anemia [30], anemia due to gastrointestinal hemorrhage [31], and myelodysplastic syndrome, respectively [24].

### 2.4. Research Regions

The studies were conducted across three countries, with the majority of studies conducted in China (n = 10/16, 62.5%) [18,19,20,25,26,27,28,29,32,33]. Two case studies were conducted in Korea (n = 2/16, 12.5%) [30,31], and two controlled clinical trials, one case study, and one case series were conducted in Japan (n = 4/16, 25%) [21,22,23,24].

### 2.5. Demographic Characteristics of Study Participants

#### 2.5.1. Participant Number, Sex, and Age

After excluding literature reviews, 863 participants were involved in the included studies (Table 2). Among these, 112 patients were included in case reports and case series, while 751 patients participated in RCTs or controlled clinical trials. Yao et al. (2012) did not report the sex of the participants. Of the remaining 809 participants in the other studies, 425 (52.53%) were male and 384 (47.47%) were female. Patient ages ranged from 18 to 67 years, with one study not mentioning the age [25]. Interventional studies expressed age as a mean value. 

#### 2.5.2. Types of Anemia

Classified by the type of anemia, there were seven studies on post-operative anemia [18,19,23,27,29,32,33], four studies on CKD anemia [22,25,26,30], two studies on gastrointestinal bleeding anemia [28,31], one study on hemolytic anemia [21], one study on myelodysplastic syndrome [24], and one study on an unspecified type of anemia [20]. Among the nine RCTs, six addressed post-operative anemia [18,19,27,29,32,33], two focused on CKD anemia [25,26], and one investigated gastrointestinal bleeding anemia [28]. There was one case report each for CKD anemia [30], gastrointestinal bleeding anemia [30], and myelodysplastic syndrome [24]. In controlled clinical trials, there was one case each of post-operative anemia [23] and CKD anemia [22]. The case series included one of hemolytic anemia [21] and one of unknown anemia [20] (Figure 2, Table 3).

#### 2.5.3. Types of Treatment and Medication

Research on post-operative anemia focused on orthopedic surgeries, with five studies related to hip surgery [19,23,27,32,33] and two studies related to knee surgery [18,29]. Blood transfusions were administered in the research on myelodysplastic syndrome [24]. In four studies on CKD anemia, erythropoietin was used in all cases [25,26,28,30]. Iron supplements [18,20,25,26,31], folic acid [18,20,25,26,28], and vitamin B12 [20,25,26,28] were concurrently administered. In two case reports, acupuncture was performed alongside the administration of SDT [30,31] (Table 4).

### 2.6. Details in Intervention

#### 2.6.1. Dosage, Frequency, and Treatment Period of Herbal Medicine

In this context, “dosage” refers to the total combined quantity of the listed herbal medicines administered in a unit. For example, administering one dosage twice daily indicates that the specified amount is divided into two separate administrations within a 24 h period. Regarding the dosage (Table 5), when the composition was considered as one dosage unit, two case reports indicated two doses per day [30,31]. In nine RCTs conducted in China, as well as in two controlled clinical trials and a case series from Japan, the prescribed dosage was one dose per day [18,19,21,22,23,25,26,27,28,29,32,33]. One case report from Japan did not specify the dosage [24].

Regarding the frequency of SDT administration investigated in studies, it was not described in two case series [20,21], one case report [24], and two controlled clinical trials [22,23]. It was described as twice per day in nine RCTs [18,19,25,26,27,28,29,32,33] and three times per day in two case reports [30,31]. 

The treatment duration across the studies varied from 1 day to 4 years. When considering only interventional studies, the treatment duration varied from 1 day to 3 months [18,19,22,23,25,26,27,28,29,32,33]. The most common treatment duration was 14 days [18,19,21,27], and six of the 16 studies had a treatment duration of 14–21 days [18,19,21,23,27,30].

#### 2.6.2. Composition of SDT and Herbal Medicines Included in Its Variants

In ten studies [18,19,20,21,22,23,28,29,30,33], the original SDT was employed, whereas six studies [24,25,26,27,31,32] applied variants of SDT. The herbal medicines added to the original SDT in these variants included Zingiberis Rhizoma Crudus, Jujubae Fructus, Pinelliae Rhizoma, Citri Pericarpium, Rhei Radix et Rhizoma, Codonopsis Pilosulae Radix, Selaginellae herba, Coicis Semen, Salviae Radix, Polygonati Rhizoma, Reynoutriae Multiflorae Radix, Ligustri Fructus, Eucommia ulmoides Oliver, and Oryzae Gluten. Among these, Rhei Radix et Rhizoma [28,29] and Codonopsis Pilosulae Radix [30,34] were the most frequently used, each appearing in two studies, whereas the other herbs were used only once. Additionally, Codonopsis Pilosulae Radix was used in two studies that reported the use of the original SDT formula [19,33]. The dosage of each herb varied widely, ranging from 1.5 g to 50 g.

Among the three studies addressing CKD anemia, two RCTs [25,26] included the addition of Rhei Radix et Rhizoma. For post-operative anemia, Codonopsis Pilosulae Radix was added in three [19,27,32] out of six RCTs, making it the most frequently added herbal medicine. Other herbal additives were added or removed according to the patient’s specific condition.

### 2.7. Evaluation Methods

#### Evaluation Tools

The evaluation tools are listed in order of frequency of use as follows: Hemoglobin levels were analyzed in all 16 studies. This was followed by hematocrit (Hct) in seven studies [18,25,26,27,31,32,33], RBC count in four studies [18,24,31,33], platelet count in three studies [18,24,28], and white blood cell (WBC) count in two studies [24,31]. In addition to Hb and Hct levels and RBC, platelet, and WBC counts, other blood tests included fasting blood glucose [32], reticulocyte [28], D-dimer [33], interleukin-6, tumor necrosis factor-alpha, C-reactive protein [18], serum creatinine, AST, ALT, sodium, and potassium levels [23]. In cases of post-operative anemia, blood loss [23,27,32,33] and transfusion volume were assessed [18,27,33].

### 2.8. Treatment Outcomes

In nine RCTs [18,19,25,26,27,28,29,32,33], hemoglobin levels were significantly improved in the patients compared with those in the control groups. Two controlled clinical trials [22,23] also demonstrated significant improvements in hemoglobin levels of patients compared with those of the controls. In three case reports [24,30,31], hemoglobin levels were restored to normal ranges in patients. Additionally, one case series [21] reported a significant reduction in hemoglobin levels in the treatment group.

Among the seven studies that measured Hct levels [18,25,26,27,31,32,33], six with control groups [18,25,26,27,32,33] showed significant improvements in Hct levels, and one case report [31] indicated a return to the normal range.

All studies focusing on post-operative anemia [18,19,23,27,29,32,33] reported significant improvements in hemoglobin levels. Additional significant improvements were observed in visual analog scale VAS scores, patient satisfaction, total blood loss, hidden blood loss, length of hospital stay, fracture healing time, post-operative drainage volume, transfusion rate, and transfusion volume.

Among the studies targeting CKD anemia, in three controlled studies [22,25,26], both hemoglobin and Hct levels showed significant increase, accompanied by a reduction in C-reactive protein levels and an improved total effective rate. A case report noted improvements in fatigue severity [30].

In a single case report of anemia following gastrointestinal bleeding, hemoglobin, Hct, RBC, and WBC levels were restored and VAS scores decreased. In one RCT [28], significant increases were observed in hemoglobin levels and reticulocyte, platelet, and WBC counts.

For myelodysplastic syndrome, a single case report [24] documented significant increases in hemoglobin and platelet levels.

Finally, in a case series focusing on hemolytic anemia [21], the reduction in hemoglobin levels was significantly less pronounced, and the requirement for dose reduction or discontinuation of ribavirin, an anemia-inducing treatment drug, was lower.

### 2.9. Safety

Adverse events were reported in four studies [18,19,28,33]. The total incidence of adverse events was lower in the treatment groups than in the control groups in each study, with rates of 4%/34%, 13.64%/30.43%, 30%/47.5%, and 5.56%/27.78%, respectively.

## 3. Discussion

This study aimed to investigate the status of studies applying SDT to various types of anemia. The analysis was conducted based on four detailed questions: (1) what type of studies have been conducted? (2) what types of anemia has SDT been used for? (3) what improvements did SDT make when applied to patients? and (4) what is the level of evidence supporting the efficacy of herbal medicines?

### 3.1. Research Status

All the 16 studies included in the analysis were conducted in East Asian countries, specifically South Korea, China, and Japan. Although a worldwide database was selected for the search, the papers selected for analysis were limited to East Asia, likely because research utilizing SDT was predominantly published in this region. Among the 16 studies that applied SDT to anemia, nine were RCTs, constituting the highest proportion (56.25%), and all RCTs were conducted in China. Anemia can be easily diagnosed through blood tests that measure hemoglobin and Hct levels. Therefore, it is plausible that conducting RCTs for anemia is easier than for other diseases. All RCTs compared a control group undergoing conventional Western medical treatment for anemia with an experimental group receiving the same conventional treatment combined with SDT, regardless of the etiology of anemia.

Post-operative anemia accounted for the highest proportion of the analyzed studies, with six RCTs and one controlled clinical trial representing a significant number of high evidence level studies. In contrast, CKD anemia had two RCTs and one case report, gastrointestinal bleeding anemia had one RCT and one case report, hemolytic anemia had one case series, and myelodysplastic syndrome had one case report, indicating relatively fewer studies.

### 3.2. Types of Anemia

The most common type of anemia treated with SDT was post-operative anemia (43.75%). Studies on post-operative anemia showed significant improvements in hemoglobin and RBC levels and reductions in intra-operative transfusion requirements and hidden blood loss post-surgery with SDT administration. Intra-operative transfusions can be used to correct anemia or anticoagulant imbalances during and after surgery; surgery itself can increase the risk of venous thromboembolism (VTE), which can be dose-dependent on the amount of transfused blood [35]. Therefore, reducing the amount of transfusion through SDT administration may help lower the risk of VTE. The second most frequently studied type was CKD anemia, and recombinant human erythropoietin was chosen as the basic treatment in all the included studies.

### 3.3. Significance of SDT Treatment

All studies included in the analysis examined changes in hemoglobin levels. All 11 interventional studies (nine RCTs and two controlled clinical trials) showed significant increases in hemoglobin levels in the treatment group compared with the control group. Additionally, three case reports and one case series demonstrated a return to normal hemoglobin levels, indicating that SDT administration increased hemoglobin levels in different types of anemia, such as post-operative anemia, CKD anemia, myelodysplastic syndrome, gastrointestinal bleeding anemia, and hemolytic anemia. Studies on post-operative anemia reported significant increases in hemoglobin levels and overall patient condition improvements, including VAS pain scores, patient satisfaction, total blood loss, hidden blood loss, length of hospital stay, fracture healing time, post-operative drainage volume, transfusion rate, and transfusion volume, suggesting that SDT is effective in improving anemia during the post-operative recovery process. In studies on CKD anemia, significant increases in Hb and Hct levels were observed, indicating that SDT may contribute to the improvement of anemia caused by reduced kidney function. Furthermore, studies on anemia following gastrointestinal bleeding showed recovery in Hb, Hct, RBC, and WBC levels and a decrease in VAS scores, whereas studies on myelodysplastic syndrome and hemolytic anemia reported significant therapeutic effects. This suggests that SDT may be useful for treating anemia under various pathological conditions.

### 3.4. Proposed Treatment Mechanisms for SDT in Anemia 

The studies included in the analysis examined the effects of various herbal medicines of SDT on the hematopoietic system, bone marrow, and erythrocytes. Experimental research showed that Rehmanniae Radix preparata, a component of SDT, stimulates the hematopoietic system and increases the secretion of various hematopoietic factors [34]. Additionally, SDT was shown to be effective in restoring hematopoietic function in damaged bone marrow [36]. Another study highlighted that SDT contains substances that promote the growth of hematopoietic stem cells and play a crucial role in bone marrow stimulation, including essential fatty acids [37]. Research on Palmultang (Ba Bao Tang in Chinese and Hachimotsuto in Japanese), which includes components of SDT, demonstrated that it stimulates the transcription of erythropoietin mRNA in the liver and kidneys and enhances serum erythropoietin expression [38]. Moreover, extracts of Paeoniae Radix alba and Astragali Radix, which are constituents of SDT, were reported to increase hemoglobin levels and RBC volume [39].

Based on these findings, the multi-compound formulation of SDT appears to influence the hematopoietic system by enhancing serum erythropoietin expression and stimulating the bone marrow, which could benefit patients with anemia. However, to elucidate these effects more clearly, additional well-designed clinical and experimental studies are required.

### 3.5. Limitations of This Study and Suggestions for Further Studies

Based on the analysis of the reviewed studies, applying SDT for anemia generally improved hemoglobin and Hct levels in anemia patients. However, the amount of the included herbs varied across studies, and the dosages also differed, with some studies using one dose and others using two doses. Additionally, in terms of herbal composition, some studies addressing CKD anemia included the addition of Rhei Radix et Rhizoma, whereas studies on post-operative anemia often added Codonopsis Pilosulae Radix to the SDT. This suggests the need to standardize SDT prescription and format based on the type of anemia.

The analysis showed that hemoglobin and Hct levels significantly improved in controlled studies and returned to normal ranges in case reports, indicating their reliability as measures. Therefore, it is essential to include blood tests for assessing hemoglobin and Hct levels in study designs, and efforts should be made to standardize these measures. The treatment duration in the included studies varied from 1 day to 4 years, but the most commonly used treatment period was 14 days, with 6 out of 16 studies having a duration of 14 to 21 days. Therefore, a 2 to 3 weeks administration period seems to be the most appropriate; however, additional studies are required to confirm the treatment period. 

Finally, although all studies demonstrated improvements in hemoglobin levels and significant enhancements in outcome measures, there were limitations due to small sample sizes, varied follow-up periods, the risk of publication bias, and heterogeneous study designs. Therefore, once well-controlled RCTs and other clinical studies are conducted, systematic reviews and meta-analyses with higher levels of evidence will be necessary to validate the effects of SDT on each type of anemia.

## 4. Materials and Methods

In this study, a scoping review design was adopted to investigate the status of studies that used SDT for anemia. A scoping review is a research approach that methodically searches for, collects, and synthesizes available knowledge to outline essential concepts, evidence types, and gaps in relevant research areas [40]. This method is useful for obtaining a broad understanding of the current state of research on a particular topic by examining the scope and characteristics of available studies [40,41].

This study adhered to the methodology outlined by Arksey and O’Malley [40] and the Expanded Reporting Items for Systematic Reviews and Meta-Analyses (PRISMA-ScR) for scoping reviews. The research was executed across the following five stages: (1) establishment of research questions; (2) identification of relevant studies; (3) selection of studies; (4) organization of data; and (5) collection, summary, and reporting of results.

### 4.1. Identifying the Research Questions

Before starting the research, we posed the question: “What studies have applied SDT to anemia?” Therefore, the primary objective of this review was to analyze the existing literature on the clinical outcomes of SDT for anemia. To clarify the objectives of this review, we present the following research questions. (1) What types of studies have been conducted? (2) What types of anemia have SDT been used for? (3) What improvements did SDT make when applied to patients? (4) What is the level of evidence supporting the efficacy of herbal medicines?

### 4.2. Literature Search

The PICO framework guided the formulation of the search query, which focused on the use of SDT to treat anemia. The search specifically included clinical studies involving human participants. Studies conducted at the cellular level or utilizing animal models were excluded. The scope of the intervention was confined exclusively to the herbal formula, SDT, as it pertains to TEAM. The etiology of anemia treatment included all types. SDT variants (Gamisipjeondaebotang in Korean, Jiawei Shiquan Dabu Tang in Chinese, and Kamijuzentaihoto in Japanese), which adjust some of the ten ingredients based on the patient’s condition or symptoms, were also included in the study. Formulations, such as decoctions, pills, powders, and capsules, were included without restrictions. Studies that used non-oral administration methods, including acupoints or intravenous injections, were excluded. No specific constraints were imposed on the types of controls or outcomes considered. Eligible studies were focused exclusively on clinical research and encompassed a variety of research designs, including case reports, case series, prospective and retrospective observational studies, before-and-after studies, RCTs, integrative literature reviews, and systematic reviews. The selection was limited to studies fully published in academic journals, and unpublished manuscripts while conference abstracts were not considered.

Six electronic databases were used for the literature review: MEDLINE (PubMed), Embase, Cochrane Library, National Digital Science Library, China National Knowledge Infrastructure, and SCOPUS. In addition to the databases, relevant studies, previously identified but not captured through the search strategy, were also included in the analysis. A comprehensive search of all titles, abstracts, subject headings, and studies published up to 30 June 2024, was conducted on 11 July 2024, using the specified term combinations detailed in Appendix A.

### 4.3. Literature Selection

All literature search results were organized using EndNote version 21. Two reviewers (GK and HL) independently reviewed the titles, abstracts, and full texts of the studies, and selected the appropriate literature based on their understanding of the process and purpose of this study. The second reviewer (HL) re-evaluated the literature through a primary review. Disagreements during the literature selection process were resolved through discussion between the two reviewers. If a consensus could not be reached, the final decision was made by a third reviewer (SK).

After eliminating duplicates, studies were selected based on their titles and abstracts, while adhering to the inclusion and exclusion criteria. The introductory sections of the selected studies were then reviewed to finalize the selection for inclusion.

### 4.4. Data Extraction and Schematization

The studies ultimately included were analyzed and discussed in relation to the primary research questions. To establish the basic characteristics of these studies, we compiled information on the authors, publication year, study field, and design. Each study was catalogued in a structured format that encompassed essential details about the participants, classification of diseases, treatment specifics, methods of assessment, and primary outcomes, facilitating an understanding of the participant demographics and interventions, as well as the impact of the treatments.

The data gathered primarily pertained to observational and interventional studies, with the necessary details documented in systematic reviews. In general, all information was transcribed exactly as it appeared in the source material. Authorship was noted using the surname of the first author, publication year was noted as the year of entry into the bibliographic database, and field of study was defined as the discipline of the first author’s affiliated institution. The study designs were categorized according to the methodologies outlined in the Public Health Guidance (2012) issued by the National Institute for Health and Care Excellence [42]. Documents were categorized based on established criteria; if the methodology was not explicitly defined, the most suitable classification was determined based on the document’s content. The research designs were broadly segmented into observational, interventional, and literature studies.

For publication year, research design, and research area, a table was compiled, listing the number of documents and their proportion relative to the total document count. Excel 2019 was used to create a chart to examine the distribution of study designs over time. Using Python 3.10.6, a sunburst chart was generated, the inner circle of which categorizes the studies, whereas the outer circle details the specific diseases addressed by these studies.

Furthermore, basic details about the study participants were outlined, including total participant count, sex distribution, age range, duration of treatment, and specific type of anemia. If only the sex ratio was available, approximate numbers were deduced by applying the sex ratio to the total number of participants. Treatment duration was expressed in days or months. In cases where a control group was present, details for both the treatment and control groups were provided separately. Data not specified were labeled as “Not mentioned”, and the diagnostic criteria for anemia were explicitly described as per the documents.

For treatment details, the records included both herbal medicine treatments and supplementary therapies used in conjunction with herbal medicine. The specific name of the herbal prescription, its composition, administration methods (frequency and dosage per day), and total duration of treatment were documented. If frequency or dosage was not mentioned, it was labeled as “Not mentioned.” If the treatment duration was absent, it was estimated in days from the visit to the treatment end date. In cases where the prescription composition was not detailed, it was recorded according to the original source, with reference to the Korea Pharmaceutical Information Center (www.health.kr). Concurrent treatments, including conventional therapies and other TEAM therapies, such as acupuncture, cupping, and moxibustion, were also summarized. 

The evaluation methods employed in each study were analyzed based on the descriptions in the text. Observational studies concentrated on documenting clinical progression. Interventional studies rigorously examined the evaluation methods specified in the research design.

The main findings were derived from the treatment outcomes emphasized in the Discussion and Conclusion Sections of the studies. Improvements were also noted if pre-treatment results for the evaluation tools were available. In cases of multiple assessments, both the initial and final results were documented.

## Figures and Tables

**Figure 1 pharmaceuticals-17-01192-f001:**
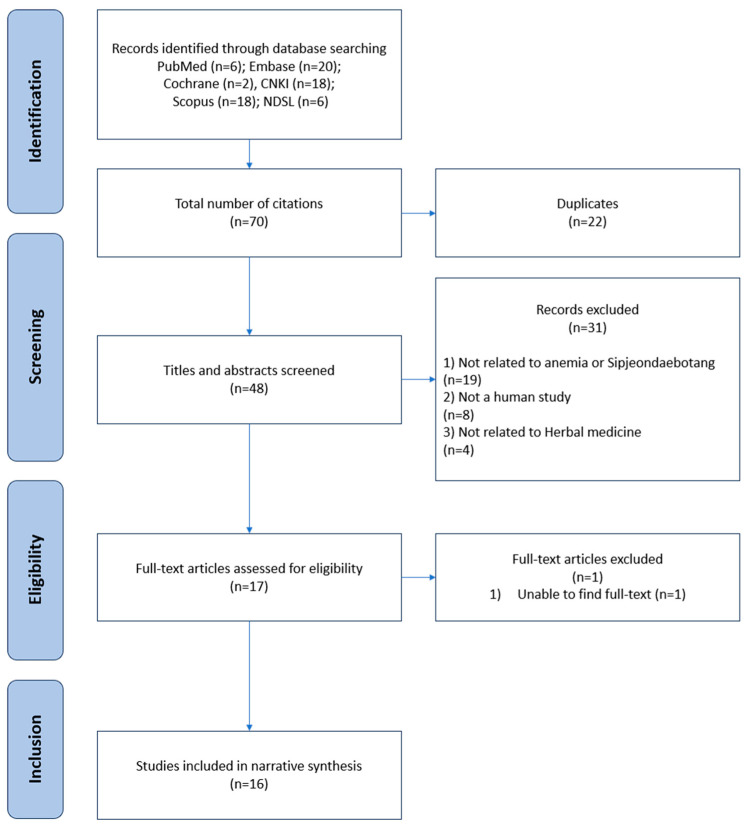
Study flow chart.

**Figure 2 pharmaceuticals-17-01192-f002:**
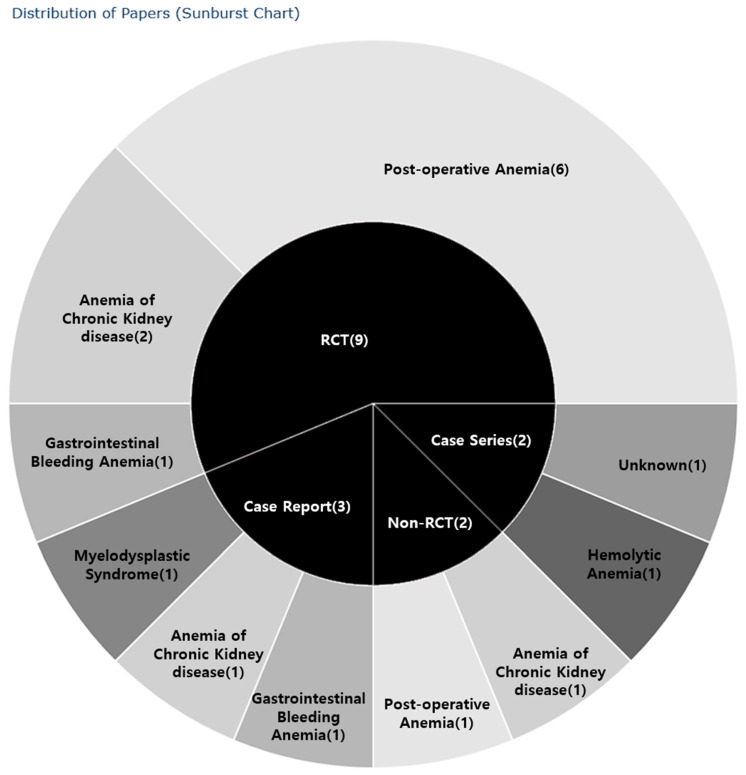
Distribution of studies by type of research. RCT—randomized controlled trial.

**Table 1 pharmaceuticals-17-01192-t001:** The 10 Herbal medicines that constitute Sipjeondaebotang.

Herbal Medicine	Scientific Name (Family)
Ginseng Radix	*Panax ginseng* C. A. Meyer (Araliaceae)
Astragali Radix	*Astragalus membranaceus* Bunge (Fabaceae)
Hoelen alba	*Poria cocos* Wolf (Polyporaceae)
Atractylodis Rhizoma Alba	*Atractylodes japonica* Koidzumi (Asteraceae)
Glycyrrhizae Radix	*Glycyrrhiza uralensis* Fischer (Fabaceae)
Rehmanniae Radix preparate *	*Rehmannia glutinosa* (Liboschitz ex Steudel) (Plantaginaceae)
Angelicae gigantis Radix	*Angelica gigas* Nakai (Apiaceae)
Cnidii Rhizoma	*Cnidium officinale* Makino (Apiaceae)
Paeoniae Radix alba	*Paeonia lactiflora* Pallas (Paeoniaceae)
Cinnamomi Cortex	*Cinnamomum cassia* Presl (Lauraceae)

* *Rehmanniae Radix preparata* is the steamed and dried root of *Rehmannia glutinosa* (Liboschitz ex Steudel).

**Table 2 pharmaceuticals-17-01192-t002:** General characteristics of the included studies (n = 16).

Variables	Categories	N (%)
Publication year	1990–1999	1 (6.25%)
	2000–2009	3 (18.75%)
	2010–2024	12 (75%)
Research Methodology	Case series	2 (12.5%)
	Case report	3 (18.75%)
	Randomized controlled trial	9 (56.25)
	Non-randomized controlled trial	2 (12.5%)
Location	China	10 (62.5%)
	South Korea	2 (12.5%)
	Japan	4 (25%)

**Table 3 pharmaceuticals-17-01192-t003:** Demographic and clinical characteristics of patients.

Location/ Design	Sample Size (Male/Female)	Age (Treatment/Control)	Types of Anemia	Author (Year)
South Korea/ Case report	1(0/1)	76	Gastrointestinal bleeding anemia	Kim (2020) [31]
South Korea/ Case report	1(1/0)	47	CKD anemia	Kim (2019) [30]
China/RCT	54	Not mentioned	CKD anemia	Yao (2012) [25]
China/RCT	120(70/50)	40.3 ± 2.8	CKD anemia	Yao (2016) [26]
China/RCT	60(31/29)	(64.21 ± 2.29 /66.21 ± 1.86)	Post-operative anemia (knee joint replacement)	Fang (2018) [29]
China/RCT	120(73/47)	71.47 ± 4.66	Post-operative anemia (proximal femur nail antirotation)	Li (2020) [32]
China/RCT	100(57/43)	(48.46 ± 3.66 /48.14 ± 3.27)	Gastrointestinal bleeding anemia	Li (2018) [28]
China/ Case series	42(23/19)	18~67	Unknown (no mention)	Wang (1999) [20]
China/RCT	45(22/23)	(66.32 ± 8.09 /63.57 ± 6.32)	Post-operative anemia (total hip arthroplasty)	Tian (2020) [33]
China/RCT	40(13/27)	65~82 (68.7 ± 5.33 /70.7 ± 4.68)	Post-operative anemia (total hip arthroplasty)	Sheng (2017) [27]
China/RCT	80(34/46)	75~95(83.6 ± 7.2) /76~93(82.4 ± 5.8)	Post-operative anemia (proximal femur nail antirotation)	Ai (2022) [19]
Japan/ Case report	1(0/1)	76	Myelodysplastic syndrome	Beinu (2011) [24]
China/RCT	72(26/46)	(67.36 ± 6.38 /67.89 ± 7.47)	Post-operative anemia (total knee arthroplasty)	Chen (2024) [18]
Japan/ Controlled clinical trial	18(0/18)	(49.4 ± 9.4 /52.1 ± 3.1)	Post-operative anemia (total hip arthroplasty or rotational acetabular osteotomy)	Kishida (2009) [23]
Japan/ Controlled clinical trial	42(33/9)	(67 ± 12/62 ± 12)	CKD anemia	Nakamoto (2008) [22]
Japan/ Case series	67(42/25)	(54.8 ± 9.0/54.6 ± 7.8)	Hemolytic anemia	Sho (2004) [21]

**Table 4 pharmaceuticals-17-01192-t004:** Summary of the included studies.

Intervention	Other Treatment	Other Medications	Outcomes	Significant Findings	Adverse Events	References
SDT variants 1	Acupuncture	Beta-blocker Vasodilator Analgesics Antiemetic Iron supplements Statin Proton pump inhibitor	Hb RBC WBC Hct Intensity of dizziness (VAS)	The major blood parameters returned to normal ranges Hb 8.8 g/dL → 11.8 g/dL RBC 2.68 × 10^6^/μL → 3.86 × 10^6^/μL WBC 2.21 × 10^3^/μL → 3.16 × 10^3^/μL Hct 26% → 35.9% The patient’s subjective symptom of dizziness VAS 5 → 1	Not mentioned	Kim (2020) [31]
SDT	Acupuncture	Antacid Amino acid rHuEPO	Hb Fatigue severity scale(FSS)	The patient’s hemoglobin levels improved after taking the medicine Hb 9.6 g/dL → 12.7 g/dL The patient’s fatigue severity scale (FSS) was improved 35 → 15	Not mentioned	Kim (2019) [30]
SDT variants 2	(-)	rHuEPO Iron Folic acid Vitamin B12	Hb Hct	The patient’s hemoglobin and hematocrit level of the treatment group were significantly increased Treatment group: Hb 64.68 ± 9.11 → 105.82 ± 13.66 Hct 20.87 ± 2.18 → 32.51 ± 3.11 Control group: Hb 64.42 ± 7.50 → 95.23 ± 12.50 Hct 20.47 ± 2.23 → 28.57 ± 2.68	Not mentioned	Yao (2012) [25]
SDT variants 3	(-)	rHuEPO Iron Folic acid Vitamin B12	Treatment effective rate Hb Hct	Total effective rate of treatment group was significantly higher and the patient’s hemoglobin and hematocrit level of the treatment group were significantly increased Treatment group: Total effective rate Hb 62.46 ± 12.64 → 106.21 ± 11.61 Hct 21.52 ± 3.20 → 31.51 ± 4.21 Control group Total effective rate 81.67 Hb 63.13 ± 14.01 → 87.07 ± 12.21 Hct 20.17 ± 3.27 → 25.81 ± 4.67	Not mentioned	Yao (2016) [26]
SDT	Knee joint replacement	Tranexamic Acid Sodium Chloride Morphine Epinephrine Ropivacaine	Amount of bleeding and drainage Hb Knee resting pain score(VAS)	The Hb in the treatment group was significantly improved compared to the control group The knee resting pain score (VAS) of treatment group was significantly lower than that of control group Treatment group: The patient satisfaction of treatment group was significantly higher than that of control group Treatment group: Hb 104.36 ± 8.83 → 117.72 ± 6.07 VAS 6.31 ± 0.52 Satisfaction 96.0 Control group: Hb 107.83 ± 9.08 → 109.62 ± 5.38 VAS 8.02 ± 0.81 Satisfaction 82.0	Not mentioned	Fang (2018) [29]
SDT variants 4	Proximal femur nail antirotation	Not mentioned	Hb Hct Hidden blood loss Fasting blood glucose level Hospitalization and fracture healing times Hip joint function recovery rates	The Hb, Hct was significantly improved and hidden blood loss was reduced, and hospitalization and fracture healing time were shorter than control group. Treatment group Hb 122.85 ± 15.24 → 100.85 ± 13.28 Hct 33.65 ± 5.41 → 27.96 ± 3.25 Hidden blood loss 245.74 ± 88.26 → 188.26 ± 70.12 Hospitalization time 8.72 ± 2.37 Fracture healing time 12.65 ± 1.34 Control group Hb 120.96 ± 16.31 → 90.41 ± 11.27 Hct 32.98 ± 6.13 → 25.47 ± 4.52 Hidden blood loss 288.67 ± 90.94 → 245.94 ± 75.36 Hospitalization time 15.12 ± 3.41 Fracture healing time 15.27 ± 5.78	Not mentioned	Li (2020) [32]
SDT	No treatment	Erythropoietin Vitamin B12 Folic acid Somatostatin Pantoprazole sodium	Hb Reticulocytes Platelets White blood cells	The Hb was significantly improved and reticulocytes, platelets, white blood cell counts were significantly increased. Treatment group Hb 42.86 ± 5.41 → 95.47 ± 6.98 Reticulocytes 0.95 ± 0.25 → 2.48 ± 1.63 Platelets: 28.95 ± 5.42 → 59.74 ± 9.85 White blood cells: 2.44 ± 0.52 → 4.85 ± 1.6	Total adverse event rate Treatment group: 4.00% Control group: 34.00%	Li (2018) [28]
SDT	No treatment	Iron Folic acid Vitamin B12	Hb	The total effect rates were 95.2%	Not mentioned	Wang (1999) [20]
SDT	Total hip arthroplasty	Tranexamic acid	RBC Hb Hct D-dimer Total blood loss Patient blood volume Dominant blood loss Hidden blood loss Transfusion rate and average transfusion volume Harris hip score Traditional Chinese medicine (TCM) Syndrome score and efficacy	The treatment group’s RBC, Hb, Hct were significantly improved and D-dimer, total blood loss, hidden blood loss were significantly lower than control group. The Harris hip score and TCM syndrome efficacy were significantly higher in treatment group.	Total adverse event rate Treatment group: 13.64% Control group: 30.43%	Tian (2020) [33]
SDT variants 5	Total hip arthroplasty	Not mentioned	Hb Hct Total blood loss Intra-operative blood loss Post-operative drainage Hidden blood loss Transfusion rate	The treatment group’s Hb and Hct were significantly improved and total, post-operative drainage, hidden blood loss were significantly reduced. Transfusion rate was also lower in treatment group. Treatment group Hb 131.5 ± 15.1 → 93.5 ± 17.9 Hct 0.42 ± 0.05 → 0.29 ± 0.05 Post-operative drainage 456.2 ± 147.3 Hidden blood loss 796.7 ± 354.6 Transfusion rate 15% Control group Hb 130.5 ± 13.9 → 83.5 ± 9.6 Hct 0.41 ± 0.06 → 0.24 ± 0.04 Post-operative drainage 612.4 ± 161.5 Hidden blood loss 1064.6 ± 335.7 Transfusion rate 35%	No adverse effect	Sheng (2017) [27]
SDT	Proximal femur nail antirotation	Antibiotics Ribaroxaban Albumin	Hb Serum albumin Transfusion rate Transfusion volume Serum albumin infusion rate Serum albumin infusion volume	The treatment group’s Hb level was significant increased and transfusion rates and volume were significantly lower than control group. Treatment group Hb 115.24± 6.13 → 80.45 ± 9.48 Transfusion rate 20.0% Transfusion volume 1.94 ± 0.44 Control group Hb 116.43 ± 3.64 → 71.81 ± 12.43 Transfusion rate 40.0% Transfusion volume 3.86 ± 2.62	Total adverse event rate Treatment group: 30% Control group: 47.5%	Ai (2022) [19]
① SDT ② SDT variants 6 ③ SDT variants 7	Transfusion	Vitamin K2 Hwangigunjung-tang Samul-tang	WBC RBC Hb Platelet	After adding malt sugar to Sipjeondaebotang, hemoglobin and platelet levels significantly increased. Bone marrow findings showed a reduction in erythroblasts.	No adverse effect	Beinu (2011) [24]
SDT	Total knee arthroplasty	Iron sucrose injection	Hb RBC Hct Platelet Interleukin-6 Tumor necrosis factor-alpha C-reactive protein Hospital for special surgery knee score(HSS) Visual analogue scale (VAS) Perioperative blood loss and transfusion rate Post-operative complication rate	The effective rate of the treatment group and the HSS score were significantly higher than that of the control group, and the levels of Hb, RBC, Hct were significantly increased. The levels of protein and CRP and the VAS score, and the perioperative blood transfusion rate were significantly lower than that in the control group. Treatment group Effective rate 91.7% Hb 130.44 ± 7.13 → 121.39 ± 6.05 RBC 4.30 ± 0.25 → 3.96 ± 0.19 Hct 39.88 ± 3.30 IL-6 97.52 ± 5.89 → 31.80 ± 2.58 TNF-a 17.51 ± 2.18 → 7.16 ± 1.11 CRP 26.87 ± 3.38 → 11.70 ± 1.87 HSS score 61.92 ± 2.87 → 85.86 ± 2.65 VAS score 6.87 ± 1.10 → 1.17 ± 0.43 Perioperative transfusion rate 8.33% Control group Hb 130.44 ± 7.13 → 121.39 ± 6.05 RBC 4.30 ± 0.25 → 3.96 ± 0.19 Hct 39.88 ± 3.30 IL-6 98.22 ± 6.08 → 34.52 ± 2.79 TNF-a 17.32 ± 2.39 → 8.31 ± 1.30 CRP 26.34 ± 5.53 → 13.37 ± 2.11 Perioperative transfusion rate 27.78% HSS score 61.53 ± 3.08 → 84.67 ± 2.08 VAS score 6.81 ± 1.12 → 1.93 ± 0.47	Total adverse event rate Treatment group: 5.56% Control group: 27.78%	Chen (2024) [18]
SDT	Total hip arthroplasty or rotational acetabular osteotomy Autologous donation	Not mentioned	Hb blood loss post-operative recovery rate laboratory tests (serum creatinine, AST, ALT, Na, K)	The Hb levels were significantly improved during the preoperative period. At the last autologous donation, the levels of Hb were higher in treatment group than control group. Treatment group Reduction of Hb at last autologous donation 0.7 g/dL Control group Reduction of Hb at last autologous donation 1.5 g/dL	No adverse effect	Kishida (2009) [23]
SDT	Not mentioned	Erythropoietin	Hb CRP	The Hb levels of treatment group were significantly increased, and the CRP levels were significantly decreased compared to the control group. Treatment group Hb 8.4 ± 1.10 → 9.5 ± 1.3 CRP 1.4 ± 1.7 → 0.6 ± 0.8 Control group Hb 8.3 ± 0.7 → 8.5 ± 0.5 CRP did not significantly change	No adverse effect in treatment group	Nakamoto (2008) [22]
SDT	Not mentioned	Interferon plus ribavirin(IFN/Rib)	Hb ALT	The Hb levels of treatment group showed significant reduction in HB decrease, and requirement for ribavirin dose reduction or withdrawal were significantly lower than that of the control group. Treatment group Requirement for ribavirin dose reduction/withdrawal 13% (4/32) Control group Requirement for ribavirin dose reduction/withdrawal 13% (4/32)	Not mentioned	Sho (2004) [21]

SDT, Sipjeondaebotang. Hb, Hemoglobin, RBC, Red blood cell. WBC, White blood cell. Hct, Hematocrit. rHuEPO, Recombinant human erythropoietin. AST, aspartate transaminase. ALT, Alanine aminotransferase. CRP, C-reactive protein.

**Table 5 pharmaceuticals-17-01192-t005:** Summary of the ingredients, dosage, frequency, and treatment duration of the included studies. The family name of the plant of origin of the herbal medicine is in parentheses.

Intervention	Composition	Dosage	Frequency	Treatment Duration	References
SDT variants 1	*Panax ginseng* C. A. Meyer (Araliaceae) 6 g, *Astragalus membranaceus* Bunge (Fabaceae) 3 g, *Wolfiporia extensa* (Peck) Ginns (as *Poria cocos* Wolf) (Polyporaceae) 6 g, *Atractylodes japonica* Koidzumi (Asteraceae) 6 g, *Glycyrrhiza uralensis* Fischer (Fabaceae) 6 g, *Rehmannia glutinosa* (Liboschitz ex Steudel) (Plantaginaceae) 3 g, *Angelica gigas* Nakai (Apiaceae) 3 g, *Cnidium officinale* Makino (Apiaceae) 3 g, *Paeonia lactiflora* Pallas (Paeoniaceae) 3 g, *Cinnamomum cassia* Presl (Lauraceae) 3 g, *Zingiber officinale* Roscoe (Zingiberaceae) 6 g, *Ziziphus jujuba* Mill. (Rhamnaceae) 4 g, *Pinellia ternata* (Thunb.) Makino ex Breitenbach (Araceae) 4 g, *Citrus reticulata* Blanco (Rutaceae) 4 g.	2 doses per day	Three times per day p.o	41 days	Kim (2020) [31]
SDT	*P. ginseng* 6 g, *A. membranaceus* 6 g, *W. extensa* 5 g, *A. japonica* 5 g, *G. uralensis* 5 g, *R. glutinosa* 5 g, *A. gigas* 5 g, *C. officinale* 5 g, *P. lactiflora* 5 g, *C. cassia* 6 g, *Z. jujuba* 6 g, *Z. officinale* 6 g	2 doses per day	Three times per day p.o	16 days	Kim (2019) [30]
SDT variants 2	*P. ginseng* 9 g, *A. membranaceus* 30 g, *W. extensa* 12 g, *A. japonica* 10 g, *G. uralensis* 6 g, *R. glutinosa* 20 g, *A. gigas* 15 g, *C. officinale* 9 g, *P. lactiflora* 10 g, *C. cassia* 5 g, *Rheum palmatum* L. (Polygonaceae) 9 g.	1 dose per day	Two times per day p.o	3 months	Yao (2012) [25]
SDT variants 3	*A. membranaceus* 30 g, *W. extensa* 12 g, *A. japonica* 10 g, *G. uralensis* 6 g, *R. glutinosa* 20 g, *A. gigas* 15 g, *C. officinale* 9 g, *P. lactiflora* 10 g, *C. cassia* 5 g, *R. palmatum* 9 g.	1 dose per day	Two times per day p.o	3 months	Yao (2016) [26]
SDT	*A. membranaceus* 50 g, *W. extensa* 10 g, *A. japonica* 13 g, *G. uralensis* 6 g, *A. gigas* 10 g, *C. officinale* 8 g, *P. lactiflora* 13 g, *Equus asinus* L. (Equidae) *(as Asini Corri* Colla) 10 g.	1 dose per day	Two times per day p.o	1 day	Fang (2018) [29]
SDT variants 4	*A. membranaceus* 15 g, *A. japonica* 15 g, *C. officinale* 10 g, *R. glutinosa* 10 g, *A. gigas* 10 g, *W. extensa* 10 g, *P. lactiflora* 10 g, *C. cassia* 3 g, *G. uralensis* 3 g, *Codonopsis pilosula* (Franch.) Nannf. (Campanulaceae) 15 g.	1 dose per day	Two times per day p.o	3 days	Li (2020) [32]
SDT	*P. ginseng* 6 g, *A. membranaceus* 12 g, *W. extensa* 9 g, *A. japonica* 9 g, *G. uralensis* 3 g, *R. glutinosa* 12 g, *A. gigas* 9 g, *C. officinale* 6 g, *P. lactiflora* 9 g, *C. cassia* 3 g.	1 dose per day	Two times per day p.o	10 days	Li (2018) [28]
SDT	*P. ginseng* 6 g, *A. membranaceus* 20 g, *W. extensa* 15 g, *A. japonica* 20 g, *G. uralensis* 6 g, *R. glutinosa* 20 g, *A. gigas* 20 g, *C. officinale* 12 g, *P. lactiflora* 20 g, *C. cassia* 5 g, *R. glutinosa* 20 g, *Leonurus japonicus* (Lamiaceae) 15 g, *Crataegus pinnatifida* (Rosaceae) 10 g, *Triticum aestivum* L. 10 g, *Hordeum vulgare* (Poaceae) 10 g.	1 dose per day	Not mentioned	3 months~4 years (mean 18 months)	Wang (1999) [20]
SDT	*A. membranaceus* 12 g, *W. extensa* 9 g, *A. japonica* 9 g, *G. uralensis* 3 g, *R. glutinosa* 12 g, *A. gigas* 9 g, *C. officinale* 6 g, *P. lactiflora* 9 g, *C. cassia* 3 g, *C. pilosula* 9 g.	1 dose per day	Two times per day p.o	7 days	Tian (2020) [33]
SDT variants 5	*A. membranaceus* 15 g, *W. extensa* 10 g, *A. japonica* 15 g, *G. uralensis* 3 g, *R. glutinosa* 15 g, *A. gigas* 10 g, *C. officinale* 10 g, *P. lactiflora* 10 g, *C. cassia* 3 g, *C. pilosula* 15 g, *Selaginella doederleinii* (Selaginellaceae) 15 g, *Coix lacryma-jobi* (Poaceae) 30 g, *Salvia miltiorrhiza* (Lamiaceae) 20 g, *Polygonatum odoratum* (Asparagaceae) 15 g, *Polygonum multiflorum* (Polygonaceae) 15 g, *Ligustrum lucidum* (Oleaceae) 10 g, *Eucommia ulmoides* (Eucommiaceae) 15 g.	1 dose per day	Two times per day p.o	14 days	Sheng (2017) [27]
SDT	*A. membranaceus* 20 g, *W. extensa* 15 g, *A. japonica* 20 g, *G. uralensis* 3 g, *R. glutinosa* 15 g, *A. gigas* 15 g, *C. officinale* 15 g, *P. lactiflora* 15 g, *C. cassia* 3 g, *C. pilosula* 20 g.	1 dose per day	Two times per day p.o	14 days	Ai (2022) [19]
① SDT ② SDT variants 6 ③ SDT variants 7	①: *P. ginseng* 3 g, *A. membranaceus* 3 g, *W. extensa* 3 g, *A. japonica* 3 g, *G. uralensis* 1.5 g, *R. glutinosa* 3 g, *A. gigas* 5 g, *C. officinale* 3 g, *P. lactiflora* 3 g, *C. cassia* 3 g ②: in ①, *R. glutinosa, C. officinale, P. lactiflora, A. gigas* were changed to 6 g. ③: in ②, add *Oryza sativa* L. (Poaceae) 10 g	Not mentioned	Not mentioned	31 months	Beinu (2011) [24]
SDT	*P. ginseng* 6 g, *A. membranaceus* 20 g, *W. extensa* 15 g, *A. japonica* 15 g, *G. uralensis* 6 g, *R. glutinosa* 15 g, *A. gigas* 15 g, *C. officinale* 6 g, *P. lactiflora* 6 g, *C. cassia* 6 g, *Drynaria fortunei* J.Smith (Polypodiaceae) 10 g, *Pyritum* 10 g	1 dose per day	Two times per day p.o	14 days	Chen (2024) [18]
SDT	*P. ginseng* 3 g, *A. membranaceus* 3 g, *W. extensa* 3 g, *A. japonica* 3 g, *G. uralensis* 1.5 g, *R. glutinosa* 3 g, *A. gigas* 5 g, *C. officinale* 3 g, *P. lactiflora* 3 g, *C. cassia* 3 g	7.5 g per day	Not mentioned	21 days	Kishida (2009) [23]
SDT	*P. ginseng* 3 g, *A. membranaceus* 3 g, *W. extensa* 3 g, *A. japonica* 3 g, *G. uralensis* 1.5 g, *R. glutinosa* 3 g, *A. gigas* 5 g, *C. officinale* 3 g, *P. lactiflora* 3 g, *C. cassia* 3 g	7.5 g per day	Not mentioned	12 weeks	Nakamoto (2008) [22]
SDT	*P. ginseng* 3 g, *A. membranaceus* 3 g, *W. extensa* 3 g, *A. japonica* 3 g, *G. uralensis* 1.5 g, *R. glutinosa* 3 g, *A. gigas* 5 g, *C. officinale* 3 g, *P. lactiflora* 3 g, *C. cassia* 3 g	7.5 g per day	Not mentioned	2 weeks	Sho (2004) [21]

## Data Availability

The original contributions presented in the study are included in the article, further inquiries can be directed to the corresponding author.

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
