# Peer review of "Current Utilization and Research Status of Herbal Medicine Sipjeondaebotang for Anemia: A Scoping Review"

_pharmaceuticals, 2024, doi:10.3390/ph17091192_

Round 1

Reviewer 1 Report

Comments and Suggestions for Authors

This manuscript aims to map the existing evidence on the effectiveness of SDT, identify gaps in the research, and provide an overview of how this traditional herbal remedy is being used in clinical settings. Through this review, the authors seek to assess SDT's potential as an alternative treatment for anemia, particularly in cases where conventional treatments may be insufficient or carry significant side effects. While this review is well-designed and contributes valuable insights, there are still areas that need improvement.

Major points:

1.The abstract is concise and summarizes the study's objectives, methods, results, and conclusions. It effectively highlights the relevance of the review, but it could benefit from a more detailed description of the key findings and implications.

2.The introduction needs a clearer explanation of why SDT was chosen for this review, including more background on its historical and clinical use in Traditional East Asian Medicine (TEAM). Additionally, the introduction would be strengthened by a more detailed discussion of the specific types of anemia that SDT might be effective in treating.

3.The review is comprehensive, covering a wide range of studies from different regions. However, the literature review could be more critically engaged. For instance, while the paper lists various studies and their findings, it would be helpful to have a deeper analysis of the quality of these studies, such as their methodological strengths and weaknesses, potential biases, and how they contribute to the overall understanding of SDT's efficacy.

4.The results section could benefit from more critical analysis. For example, while the paper reports that SDT generally improved hemoglobin levels, it would be valuable to discuss the magnitude of these improvements, the consistency across studies, and any discrepancies in the findings.

5. The discussion appropriately synthesizes the findings, suggesting that SDT may be a promising treatment for various types of anemia. However, the discussion could be more balanced by addressing potential limitations of the studies reviewed, such as small sample sizes, short follow-up periods, or the risk of publication bias. Additionally, the implications for clinical practice could be expanded, offering more concrete recommendations for healthcare providers.

Minor,

1.The references would be better represented in English.

2.Please indicate the full name and family name for all herbs listed in Table 1.

3.It is better to put Appendix 1 as a supplementary file.

4.Table 3,4,5 should be reorganized. e.g., the references cited should be placed in the rightmost column.

Comments on the Quality of English Language

There are occasional grammatical errors and inconsistent terminology usage. A thorough proofreading and possible editing by a native English speaker would help enhance clarity, coherence, and overall language quality.

Reviewer 2 Report

Comments and Suggestions for Authors

The article investigates the utilization and effectiveness of Sipjeondaebotang (SDT), an herbal prescription, in treating various types of anemia by analyzing 863 participants. They concluded that SDT may be an effective treatment for anemia, improving hemoglobin levels and patient outcomes.

-The manuscript contains some grammar and punctuation errors. It should be checked by a native English speaker for accuracy. Some sentences may need to be rewritten for clarity. For example, in lines 78-80, the sentence can be revised as “This review aimed to comprehensively evaluate the current evidence supporting the use of SDT in treating anemia, determine its significance in clinical practice and identify knowledge gaps.”. In line 192, “he evaluation tools” replaced with “The evaluation tools”.

- In the title, I recommend the deletion of herbal medicine.

In lines 78-80, the sentence can be revised as “This review aimed to comprehensively evaluate the current evidence supporting the use of SDT in treating anemia, determine its significance in clinical practice and identify knowledge gaps.”. In line 192, “he evaluation tools” replaced with “The evaluation tools”.

 - According to line 380, six electronic databases were used for the literature review. Why wasn't the WOS database used? Are all of these studies in the database accurate?

- It was mentioned that the studies were primarily conducted in China, Korea, and Japan. However, only a few case studies have been conducted in Korea and Japan. It would have been better if only case studies from China were evaluated.

Comments on the Quality of English Language

Moderate editing of English language required.

Reviewer 3 Report

Comments and Suggestions for Authors

Comments on Manuscript: pharmaceuticals-3178253

The authors in the manuscript “Current utilization and research status of herbal medicine Sip-jeondaebotang for anemia: A scoping review” had the goal to give a comprehensive review of the current level of evidence supporting the use of the Sip-jeondaebotang, the plant mixture used in Chinese, Korean and Japanese traditional medicine in treating anemia, pointing out its significance in clinical practice, as well as identifying the knowledge gaps.

Please, correct the repetition of the words and language, checking thoroughly the whole text (page 2, line 79; page 2, line 80……etc)

Table 1 should be more informative - please give the full Latin name of the mentioned plants (the species name, author name, and Family name, and the mass range of each plant part used in the prescriptions tested in the described trials/cases…)

The species names, as well as the plant part used in the tested formulations – drug name should be in Italics. Please, check Table 5, and write correctly the mentioned plant drugs

Please, explain what “Rehmanniae Radix preparata” means (Table 1)

The clarification of what SDT represents is necessary, it is also obligatory to point out in what form traditional medicine is administrated, what “one” and “two“ doses represent, and the information in what pharmaceutical form the SDT might be prepared and applied….

When the plant part (drug) is mentioned for the first time, please, give the biological source (page 2, lines 64, 65).

Table 4, the used abbreviations should be presented in the legend of the table

In the Discussion, the authors should give the data on how the concurrent application of SDT with supplements/synthetic medicines was explained and how the SDT benefit, effects in patients with anaemia were validated.

The main problem with the paper is the uniformity of SDT formulation used for the anaemia treatment. Namely, the composition of the used formulation differs in the ratio of the plants that constitute the SDT, as well as in the plant species present in the formulation. The confirmation that all those different formulations might be considered the same “medicine” for anaemia treatment should be given. Based on the title of the manuscript, it is expected that one formulation was described. However, in the manuscript, the variation of the composition used in the clinical trials/cases is so broad that a more pointed discussion is necessary

Reviewer 4 Report

Comments and Suggestions for Authors

The text presents a scoping review on the use of Sipjeondaebotang (SDT) for treating anemia, which is promising but also raises several critical points. I would expect to see comments of the authors.

The inclusion of various study designs (RCTs, controlled clinical trials, case series, case reports) without a clear distinction in the analysis could lead to issues in drawing firm conclusions. Observational studies, particularly case reports and case series, offer limited evidence due to potential biases and lack of control groups.The inclusion of various study designs (RCTs, controlled clinical trials, case series, case reports) without a clear distinction in the analysis could lead to issues in drawing firm conclusions. Observational studies, particularly case reports and case series, offer limited evidence due to potential biases and lack of control groups.

Although the text emphasizes improvements in anemia outcomes, it does not address the quality of the studies included. The majority being observational studies (31.25%) might weaken the strength of the conclusions, as these studies generally provide lower levels of evidence.

The text mentions a total of 863 participants across 16 studies. While this number might seem substantial, individual study sizes may still be small, potentially affecting the reliability of the results. The review does not provide information on the sample size distribution across studies.

It is unclear whether the results apply to all age groups or specific populations (e.g., gender-specific outcomes). 

There are mentions of missing or unspecified details in the treatment regimens (frequency, dosage, composition), which could impact the reproducibility of the results and the validity of comparisons across studies.

The studies are concentrated in East Asian countries (China, Korea, Japan), which might limit the applicability of the findings to broader, more diverse populations. Differences in genetics, diet, and lifestyle between these regions and others might influence the effectiveness of SDT. Therefore, the paper seems to be quite local.

While the conclusion states that SDT "may be an effective treatment," it should be noted that the evidence base, as presented, does not justify a strong conclusion. The text acknowledges the need for further high-quality studies, but this caveat should be emphasized more strongly to avoid overstating the current evidence.

As given in Table 1, you need to give Latin binomial names of the plant species with their author names. You better give the plant part in parentheses. As it looks now, it is not appropriate. You also need to give proportions and how it is prepared. 

Comments on the Quality of English Language

Although there are minor errors, the English is acceptable.

Reviewer 5 Report

Comments and Suggestions for Authors

Current utilization and research status of herbal medicine Sipjeondaebotang for anemia: A scoping review

1.    Introduction

The paragraphs are loose and incoherent; It is necessary to observe and do a deep reading.

Observe the correct spelling of scientific names, as some are without the correct format;

Discussion

This study aimed to investigate the status of studies applying SDT to various types of anemia. The analysis was conducted based on four detailed questions: (1) what type of

studies have been conducted? (2) what types of anemia has SDT been used for? (3) what improvements did SDT make when applied to patients? and (4) what is the level of evidence supporting the efficacy of herbal medicines?

Check the paragraph above the discussion, this is methodology and not discussion of results.

In the article, the authors must include the future perspectives of the study.

Round 2

Reviewer 1 Report

Comments and Suggestions for Authors

After reviewing the revisions, it appears that several key points raised in the previous review have not been adequately addressed. These issues are crucial to improving the overall quality and clarity of the manuscript. It is important that all comments and suggestions are carefully considered and incorporated into the revised manuscript.

Minor issues:

1. Further revision should be made on the name and Latin name of the herbs listed in Table 1. The Latin names of medicinal herbs should generally be written in italics, as they are scientific names. Italicization is typically used when the Latin name refers to the specific genus and species of the herb. However, when the Latin name is used in a non-scientific context, or if it has become a common name rather than a strictly botanical reference, italics may not be necessary.

2. Please use Simplified Chinese in Appendix 1.

3. It is highly recommended to place the references cited in Tables 3, 4, and 5 in the rightmost column.

Comments on the Quality of English Language

Minor editing of English language required.

Reviewer 2 Report

Comments and Suggestions for Authors

The authors have adequately addressed the issues I raised and the manuscript has been modified accordingly.

Author Response

Thanks for you opinion.

Reviewer 3 Report

Comments and Suggestions for Authors

Comments on Manuscript ID pharmaceuticals-3178253

The authors improved their manuscript. However, some issues remained to be addressed. 

Page 2, Table 1, full Latin name should be provided - in addition to species name and author name, the Family name should be added. Author name and Family name should be written in regular font (NON-ITALICS)

Page 9, Table 5 - the biological sources of the mentioned drugs should be added - when mentioned for the first time, full Latin Name should be given - species name (italic), author name (noon-italics), and Family name (non Italics), but if it has been already mentioned, short name should be used, for example G. uralensis (italics)

Reviewer 4 Report

Comments and Suggestions for Authors

The paper seems to be improved. My critics were more comment-based and I think the authors replied convincingly. It might be accepted now.

Author Response

Thanks for your opinion.

Reviewer 5 Report

Comments and Suggestions for Authors

The manuscript has been improved.

Author Response

Thanks for your opinion.